# Identification of Heat-Resistant Varieties of Non-Headed Chinese Cabbage and Discovery of Heat-Resistant Physiological Mechanisms

**Jing Yu, Pengli Li, Song Tu, Ningxiao Feng, Liying Chang and Qingliang Niu ***

School of Agriculture and Biology, Shanghai Jiao Tong University, Shanghai 200240, China; 019150210006@sjtu.edu.cn (J.Y.); lipengli@sjtu.edu.cn (P.L.); 82170922@sjtu.edu.cn (S.T.); fnxalice@sjtu.edu.cn (N.F.); changly@sjtu.edu.cn (L.C.)
* Correspondence: qlniu@sjtu.edu.cn

**Abstract:** Affected by global warming, continuous high temperature has a negative impact on plant growth and development and become a major constraint to crop production. Germplasm resource identification has become a research hotspot in many fields, and it is also necessary to establish effective identification methods. In this study, twenty *Brassica rapa* varieties were selected to investigate the physiological and biochemical characteristics under heat stress, explore the relationship between physiological response and the heat resistance mechanism, and select some typical heat-resistant and heat-sensitive varieties. The effects of photosynthetic electron transfer and antioxidant pathway on the heat resistance of *Brassica rapa* were identified. These findings will provide important guidance for the physiological regulation and identification method of heat stress in plants.

**Keywords:** non-headed Chinese cabbage; heat stress; physiological mechanism; heat-resistant varieties

## 1. Introduction

*Brassica rapa* is a leafy vegetable that is widely cultivated in China. It is a member of the cruciferous family and ranks first among all vegetable consumption in China [1,2]. *Brassica rapa* prefers cold climates, and its most suitable cultivation temperature is 20~25 °C. Heat stress often leads to drought stress, reducing yield and quality. Drought, high temperatures, salinity, and other abiotic stresses have reduced crop yields worldwide. Among them, the rise in atmospheric temperatures caused by human activities has significantly affected agricultural ecosystem functioning. Based on current trends, it is estimated that if human activities continue, the earth's temperature will likely increase by 6.4 °C [3]. Sea levels will rise by 59 cm by the end of the 21st century due to melting glaciers [4]. Climate change increases the possibility of different natural disasters, such as floods, droughts, storms, hurricanes, and changes in precipitation patterns. Since agriculture is highly dependent on climate and sensitive to agro-climatic conditions, changes in temperature, humidity, and rainfall will adversely affect crop productivity [5].

Global warming has caused frequent occurrences of extreme heat around the world, aggravating damage and hindering plant growth, development and metabolism. Heat stress not only leads to abnormal growth, yellowing, and withered leaves but also stem thinning [6]. High-temperature stress destroys leaf growth and increases susceptibility to infectious diseases, which negatively impacts quality and yield.

With the rapid development of molecular biology, the heat resistance mechanism in many crops has been further elaborated. However, understanding the plants' physiological response to heat resistance is still an important prerequisite for variety identification. Understanding the physiological and biochemical reactions of plants and establishing simple and effective identification methods are still important contents of heat-resistant breeding of horticultural crops. It is worth noting that the heading character of Chinese

cabbage is a good indicator of heat resistance, but the evaluation system for heat resistance of *Brassica rapa* has not yet been established.

The purpose of this paper is to utilize physiological and biochemical indicators to establish an effective method for identifying heat-resistant varieties, which provided some information to explore the roles of other physiological attributes in heat stress tolerance in *Brassica rapa*.

## 2. Materials and Methods

### 2.1. Plant Material and Heat Treatment

Twenty *Brassica rapa* varieties with different heat tolerances were selected for the experiment. These accessions were collected and bred by the Shanghai Academy of Agricultural Sciences. Full and healthy seeds were selected for germination and sowing. Seeds of each line were grown in a control phytotron. The ambient conditions were set as follows: 16/8 h light/dark, 150 mol/$m^{-2}$/$s^{-1}$ light intensity, and a 50% relative humidity level. Fifty seedlings with uniform growth were selected for each variety for heat treatment. Heat treatments were performed when the plants grow at the three-leaf stage. The seedlings were exposed to heat stress conditions at 37 °C/27 °C (light/dark) for 7 days. During heat stress, the seedlings were irrigated daily to avoid drought stress.

### 2.2. Measurement of Morphological and Physiological Index

Plant height and width were determined by a ruler. The dry weight (DW) and fresh weights (FW) were measured using an electronic analytical balance. The total chlorophyll content was measured using a previously described method [7]. Chlorophyll was extracted in the dark using 10 mL ethanol (95%) at 4 C using a 0.2 g fresh leaf sample. The chlorophyll content was calculated using the absorbance (A) at 645 and 663 nm using 95% ethanol as the control. The shoot root ratio was calculated by the ratio of fresh or dry weight between the underground and aboveground parts of a plant. Under the heat stress treatment, leaf samples were collected and frozen in liquid nitrogen, and stored in a −80 °C super cold refrigerator. The leaf samples were collected for physiological studies. Referring to the manufacturer's instructions, the superoxide dismutase (SOD) activity, catalase (CAT) activity, ascorbic acid oxidase activity (AAO) activity and peroxidase (POD) activity, plant root dehydrogenase activity, the content of hydrogen, proline and hydroxyl radical scavenging rate were measured by using assay kits (Comin, Suzhou, China) [8]. The soluble sugar was measured by anthrone. The soluble protein content was measured according to the Bradford procedure [9]. The determining method for the relative conductance (REC) used in this study was described by Zheng et al. [10]. Rinse *Brassica rapa* leaves three times with distilled water, then absorb surface moisture, cut long strips, avoid leaf veins, take 0.1 g and soak in a tube containing 10 mL of distilled water for 12 h. Measure the conductivity of the solution (R1). Then, boil the tube in water for 30 min, cool it to room temperature, and then measure the conductivity(R2). REC = R1/R2 × 100%. The determining method of Malondialdehyde (MDA) content was described by Zhou et al. [11].

### 2.3. Determination of Gas Exchange Parameters

The photosynthetic-related parameters were calculated using the GFS-3000 measuring systems (Heinz Walz, Effeltrich, Germany). The experiment was conducted according to the manufacturer's procedures with minor modifications as described by Zhang et al. [12]. The gas exchange parameters including photosynthetic rate (Pn), transpiration rate (E), stomatal conductance (Gs), and intercellular carbon dioxide concentration (Ci) were measured. The fourth full-developed leaf from four plants per variety was used in this measurement.

### 2.4. Measurement of Electron Transport Rate, Quantum Yield and PQ Pools

Uniformly sized leaves from each plant were used for the measurement of photosynthetic system (PSII and PSI) efficiency. Before starting measurements, the plants were placed in a dark environment for 30 min. The $CO_2$ concentration was set at approximately

$400 \pm 10$ mmol.mol$^{-1}$. Different photosynthetic photon flux densities (PPFD) was set in this experiment. The light intensity gradient was 0, 2, 13, 40, 80, 114, 157, 213, 279, 358, 592, 956, 1207 and 1537 $\mu$mol m$^{-2}$ s$^{-1}$. The rapid light curves (RLCs) of photosynthetic parameters were measured by DUAL-PAM-100 measuring systems, including the photosynthetic electron transport rate [ETR(I), ETR(II)], the effective quantum yield photochemistry [Y(I),Y(II)]; the quantum yield of PSI non-photochemical energy dissipation due to the donor-side limitation [Y(ND)], the quantum yield of regulated energy dissipation in PSII [Y(NPQ)], the quantum yield of PSI nonphotochemical energy due to the acceptor-side limitation [Y(NA)] and nonregulated heat dissipation as Y(NO) were calculated [13]. The fourth fully expanded functional leaves from different pots were used for measurements [14]. The complementary area between the oxidation curve of P700 after single turnover flashes (ST, 50 ms, PQ pools being oxidized) and multiple turnover flashes (MT, 50 ms, PQ pools are fully reduced) excitation were used to calculate the functional pools sizes of intersystem electrons [15].

*2.5. Statistical Analysis*

According to plant performance, the heat damage index (HDI) is divided into five categories: Grade 0: the leaves have no obvious heat damage symptoms; Grade 1: The number of affected leaves in the plant is less than 1/3 of the total number of leaves in the plant; Grade 3: The number of affected leaves in the plant is greater than 1/3 of the total number of leaves and less than 1/2 of the total number of leaves in the plant; Grade 5: The number of affected leaves in the plant is greater than 1/2 of the total number of leaves and less than 2/3 of the total number of leaves in the plant; Grade 7: The number of damaged leaves in the plant is greater than 2/3 of the total number of leaves in the plant; HDI = $\sum$(Number of plants per grade $\times$ grade)/(Highest grade$\times$total number of plants) $\times$ 100%; Quantitative assessment was conducted on randomly selected samples from four independent biological replicates. One-way analysis of variance (ANOVA) was performed by IBM SPSS25.0 software (Armonk, NY, USA). Comparisons between the mean values were made by the least significant difference ($p \leq 0.05$). All the data contain mean values with standard error. The statistically analyzed data are shown with superscripted letters after the numbers which show significant differences. All graphs were plotted by GraphPad Prism 8 (San Diego, CA, USA). Cluster analysis was performed by the Omicshare tool.

## 3. Results

*3.1. Agronomic Traits and Heat Damage Indicators of Brassica rapa under Heat Stress*

In this study, 20 *Brassica rapa* varieties were treated with heat stress for 7 days using an artificial climate chamber. We measured physiological and biochemical indicators to clarify the relationship between these indicators and plant heat tolerance. Additionally, we screened heat-resistant (HR) and heat-sensitive (HS) *Brassica rapa* varieties. The phenotypes of heat-resistant and heat-sensitive seedlings under heat stress are shown in Figure 1. Under high-temperature treatment, the treated plants exhibited varying degrees of heat damage symptoms. HR plants showed less damage after three days of high-temperature treatment, whereas HS plants exhibited multiple thermal injuries such as leaf shrinkage, dehydration, chlorosis, spindling, and plant growth stagnation. After the seventh day of heat stress treatment, HR plants' leaves began to dehydrate and whiten, while HS plants showed wilting and death symptoms. The overall growth condition of HR plants is better than HS plants. On the first day of treatment, there were no significant differences between heat-sensitive and heat-resistant varieties. On the third day, there were significant differences in heat damage symptoms, and on the fifth day, the differences intensified. On the seventh day, most of the heat-sensitive varieties withered, whereas most of the heat-resistant varieties survived. Therefore, we obtained samples on the first, third, fifth, and seventh days to determine physiological indicators. The heat damage index reflects the degree of heat damage to plants, which is negatively correlated with plant heat tolerance. Table 1 shows the heat damage index of 20 varieties of non-heading Chinese cabbage after

7 days of high-temperature stress at the seedling stage. The higher the heat damage index, the higher the degree of plant damage under heat stress. The heat damage index was negatively correlated with plant heat tolerance. Plants are divided into HS and HR varieties according to the heat damage index (Table 1). The HR varieties are *liehuojingang*, *xinxiaqing*, *gaohuaqing*, and *jiaoyang*. HS varieties *xinlvxiu*, *aijiaohuang*, and *wuyueman* were screened for representatives of these varieties. These typical HS and HR varieties can be used as breeding materials for heat resistance breeding.

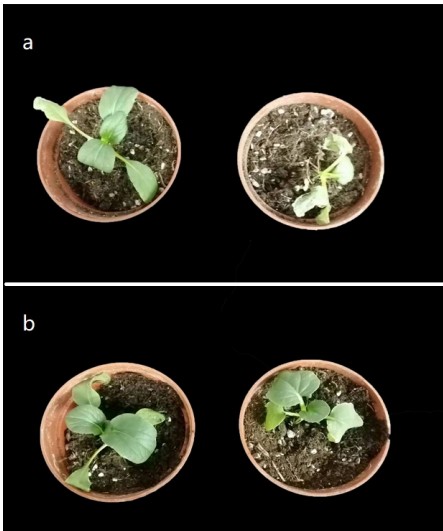

**Figure 1.** Phenotypes of HR (**left**) and HS (**right**) Brassica rapa seedlings before and after heat stress, (**a**) before and (**b**) after heat stress.

**Table 1.** The dry weight, fresh weight, and heat damage index of *Brassica rapa* cultivars with elevated heat tolerance under heat stress conditions. Different letters in a–e represent significant differences ($p < 0.05$) using one-way ANOVA. Comparisons between the mean values were made by the least significant difference.

| Variety Name | Type of Heat Tolerance | Heat Damage Index | Fresh Weight | Dry Weight |
|---|---|---|---|---|
| shanghaiqing | HS | 51.90 | 3.77 ± 0.15 bcd | 0.47 ± 0.03 b |
| wuyueman | HS | 58.67 | 4.16 ± 0.06 bcd | 0.31 ± 0.02 bcde |
| aijiaohuang | HS | 55.38 | 3.74 ± 0.09 bcd | 0.37 ± 0.02 bcde |
| baimeigui | HS | 44.75 | 4.60 ± 0.07 bc | 0.46 ± 0.03 b |
| dongguanmei | HS | 49.52 | 3.77 ± 0.22 bcd | 0.37 ± 0.02 bcde |
| gaohuaqing | HR | 37.8 | 5.12 ± 0.16 a | 0.51 ± 0.03 a |
| jiaoyang | HR | 38.00 | 4.91 ± 0.34 b | 0.47 ± 0.01 b |
| xinxiaqing | HR | 36.75 | 4.96 ± 0.16 b | 0.48 ± 0.04 b |
| Kangre605 | HR | 41.60 | 4.83 ± 0.09 b | 0.55 ± 0.03 a |
| jinfei | HS | 48.00 | 3.73 ± 0.22 bcd | 0.45 ± 0.03 b |
| liehuojingang | HR | 37.90 | 5.36 ± 0.11 a | 0.54 ± 0.03 a |
| dongxing | HS | 46.09 | 3.57 ± 0.20 bcd | 0.37 ± 0.02 bcde |
| suzhouqing | HS | 47.57 | 4.16 ± 0.53 bcd | 0.35 ± 0.02 bcde |
| jinbao | HR | 41.9 | 4.16 ± 0.14 bcd | 0.43 ± 0.03 bc |
| chibai | HS | 45.71 | 4.23 ± 0.31 bcd | 0.42 ± 0.04 bc |
| siyueman | HS | 58.00 | 3.66 ± 0.20 bcd | 0.35 ± 0.04 bcde |
| xinlvxiu | HS | 55.33 | 3.48 ± 0.20 bcd | 0.33 ± 0.11 bcde |
| xiali | HS | 46.67 | 4.30 ± 0.24 bcd | 0.34 ± 0.05 bcde |
| aoxia | HR | 41.90 | 4.22 ± 0.14 bcd | 0.44 ± 0.04 bc |
| hanxiu | HS | 50.76 | 3.91 ± 0.21 bcd | 0.32 ± 0.03 bcde |

### 3.2. Plant Growth Attributes of Non-Heading Chinese Cabbage under Heat Stress

Plants have evolved many regulatory mechanisms to acquire heat tolerance and survive under heat stress. We measured dry and fresh weight (Table 1), plant height, plant width, and leaf area (Table 2) to evaluate differences in plant biomass accumulation among HS and HR varieties under heat stress.

**Table 2.** Growth attributes of *Brassica rapa* cultivars with elevated heat tolerance under heat stress conditions. Different letters in a–e represent significant differences ($p < 0.05$) using one-way ANOVA. Comparisons between the mean values were made by the least significant difference.

| Variety Name | Plant Height | Plant Width | Leaf Area | Fv/Fm |
|---|---|---|---|---|
| shanghaiqing | 18.12 ± 1.47 a | 24.77 ± 0.61 bcd | 54.82 ± 0.63 bcde | 0.56 ± 0.03 b |
| wuyueman | 19.46 ± 1.52 bc | 21.37 ± 0.14 bcd | 43.23 ± 0.63 bcde | 0.40 ± 0.01 bc |
| aijiaohuang | 18.16 ± 0.27 a | 25.522 ± 0.29 bcd | 44.70 ± 0.59 bcde | 0.37 ± 0.00 bcd |
| baimeigui | 19.63 ± 0.30 b | 26.44 ± 0.34 bcd | 64.68 ± 1.60 a | 0.46 ± 0.01 bc |
| dongguanmei | 21.11 ± 0.53 bc | 23.94 ± 0.40 bcd | 75.35 ± 1.22 b | 0.46 ± 0.01 b |
| gaohuaqing | 23.87 ± 0.74 b | 30.85 ± 0.47 bcd | 82.74 ± 0.49 a | 0.65 ± 0.01 b |
| jiaoyang | 22.72 ± 0.40 b | 33.477 ± 0.34 a | 84.52 ± 0.07 a | 0.73 ± 0.00 b |
| xinxiaqing | 25.39 ± 0.68 bc | 23.39 ± 0.24 bcd | 85.13 ± 0.27 a | 0.78 ± 0.01 a |
| Kangre605 | 22.51 ± 0.66 a | 25.02 ± 0.42 bcd | 75.21 ± 0.42 bc | 0.53 ± 0.24 bc |
| jinfei | 18.21 ± 1.33 a | 32.95 ± 0.78 a | 72.73 ± 0.31 bc | 0.46 ± 0.01 bc |
| liehuojingang | 21.38 ± 0.38 a | 32.36 ± 0.41 a | 78.22 ± 1.09 bc | 0.75 ± 0.01 a |
| dongxing | 22.97 ± 0.62 b | 27.28 ± 0.45 bcd | 66.14 ± 2.38 bcde | 0.72 ± 0.01 a |
| suzhouqing | 20.65 ± 0.13 a | 24.91 ± 0.67 bcd | 52.88 ± 0.61 bcde | 0.62 ± 0.00 b |
| jinbao | 18.86 ± 0.83 bc | 24.10 ± 0.57 bcd | 46.80 ± 1.16 bcde | 0.50 ± 0.01 b |
| chibai | 18.98 ± 0.29 bc | 22.54 ± 0.59 bcd | 49.20 ± 0.94 bcde | 0.33 ± 0.24 bc |
| siyueman | 21.62 ± 0.81 a | 28.68 ± 0.45 bcd | 77.20 ± 0.19 bcd | 0.35 ± 0.00 bcd |
| xinlvxiu | 16.11 ± 0.67 bc | 25.30 ± 0.31 bcd | 44.79 ± 0.59 bcde | 0.36 ± 0.00 bcd |
| xiali | 19.05 ± 1.38 bc | 26.09 ± 0.39 bcd | 74.39 ± 1.30 bc | 0.54 ± 0.00 b |
| aoxia | 20.15 ± 0.54 bc | 31.18 ± 0.74 bcd | 74.37 ± 1.43 bc | 0.48 ± 0.00 bc |
| hanxiu | 20.16 ± 0.80 bc | 24.11 ± 0.15 bcd | 53.05 ± 0.53 bcde | 0.46 ± 0.01 bc |

Plants' dry and fresh weights reflect their biomass accumulation and growth rate. As shown in Table 1, heat stress considerably reduces plant growth indexes, such as fresh and dry biomass. HR plants exhibited higher dry weight accumulation compared to HS plants. Fresh and dry weights are proposed as good indicators of plant heat tolerance. We observed higher leaf area, plant width, and height in HR plants. However, the negative relationship between the heat damage index and these indicators is not significant. Table 3 assesses the root growth attributes (root dehydrogenase activity and root shoot ratio) of heat-resistant and heat-sensitive varieties under heat stress. Plant roots have multiple important functions, such as absorption, fixation, transportation, synthesis, storage, and reproduction. The plant root's dehydrogenase activity indicates excess plant root activity. This result suggests that HR plants have a more developed root system than HS plants. HR plants' stronger root activity (r = 0.684) and higher root shoot ratio (r = −0.631) help plants absorb inorganic salts and water and synthesize amino acids (Table 4).

**Table 3.** The root shoot ratio, root activity, soluble sugar, soluble protein of *Brassica rapa* cultivars with elevated heat tolerance under heat stress conditions. Different letters in a–e represent significant differences ($p < 0.05$) using one-way ANOVA. Comparisons between the mean values were made by the least significant difference.

| Variety Name | Root Shoot Ratio | Root Activity | Soluble Sugar | Soluble Protein |
|---|---|---|---|---|
| shanghaiqing | 0.044 ± 0.001 a | 33.30 ± 1.18 bcd | 2.81 ± 0.07 bc | 4.39 ± 0.05 bc |
| wuyueman | 0.038 ± 0.001 b | 38.62 ± 2.14 bcd | 2.41 ± 0.09 bc | 3.34 ± 0.02 c |
| aijiaohuang | 0.039 ± 0.001 b | 43.56 ± 0.77 bcd | 2.17 ± 0.09 bc | 3.19 ± 0.10 c |
| baimeigui | 0.039 ± 0.003 b | 41.38 ± 0.26 bcd | 2.60 ± 0.09 bc | 3.81 ± 0.04 c |
| dongguanmei | 0.040 ± 0.002 b | 51.04 ± 0.60 a | 3.35 ± 0.30 bc | 3.68 ± 0.05 c |
| gaohuaqing | 0.045 ± 0.001 a | 43.59 ± 2.50 bcd | 5.90 ± 0.11 bc | 5.80 ± 0.06 b |
| jiaoyang | 0.046 ± 0.001 a | 34.09 ± 1.15 bcd | 6.68 ± 0.22 a | 5.62 ± 0.05 b |
| xinxiaqing | 0.047 ± 0.000 a | 53.68 ± 2.94 a | 6.49 ± 0.32 b | 6.08 ± 0.09 a |
| Kangre605 | 0.042 ± 0.005 b | 36.99 ± 2.82 bcd | 5.11 ± 0.02 bc | 4.95 ± 0.05 b |
| jinfei | 0.043 ± 0.001 b | 40.80 ± 1.99 bcd | 4.67 ± 0.07 bc | 4.71 ± 0.07 b |
| liehuojingang | 0.044 ± 0.001 b | 51.34 ± 0.84 a | 6.82 ± 0.30 a | 6.31 ± 0.03 a |
| dongxing | 0.043 ± 0.001 b | 48.07 ± 2.76 b | 5.70 ± 0.16 b | 6.01 ± 0.05 a |
| suzhouqing | 0.043 ± 0.003 b | 33.17 ± 1.26 bcd | 3.27 ± 0.13 bc | 3.81 ± 0.09 c |
| jinbao | 0.043 ± 0.001 b | 32.86 ± 0.77 bcd | 3.76 ± 0.13 bc | 3.54 ± 0.04 c |
| chibai | 0.043 ± 0.001 b | 41.26 ± 1.88 bcd | 3.47 ± 0.29 cc | 3.35 ± 0.06 c |
| siyueman | 0.044 ± 0.001 b | 44.84 ± 3.56 bc | 1.80 ± 0.18 c | 3.27 ± 0.09 c |
| xinlvxiu | 0.036 ± 0.005 c | 37.33 ± 5.33 bcd | 1.98 ± 0.08 c | 3.07 ± 0.04 c |
| xiali | 0.041 ± 0.002 b | 50.71 ± 0.31 a | 4.50 ± 0.08 bc | 3.47 ± 0.09 c |
| aoxia | 0.041 ± 0.001 b | 46.90 ± 0.79 bc | 3.89 ± 0.06 bc | 3.30 ± 0.09 c |
| hanxiu | 0.041 ± 0.001 b | 43.53 ± 1.04 bcd | 3.42 ± 0.18 bc | 3.90 ± 0.02 c |

*3.3. The Chlorophyll Content, Soluble Sugar and Soluble Protein and the MDA of Non-Heading Chinese Cabbage under Heat Stress*

Plants use chlorophyll to absorb light energy for photosynthesis, which provides the necessary materials and energy for life activities. Therefore, maintaining a stable chlorophyll content is essential for the normal progress of plant photosynthesis [15]. Heat stress remarkably reduces photosynthetic pigments. As shown in Table 5, HR plants under heat stress show higher chlorophyll levels relative to HS cultivars with different ranges of Chlorophyll a (0.46–1.04 mg/g) and Chlorophyll b (0.17–0.38 mg/g). The differences between varieties are more significant in terms of Chlorophyll a content. Previous studies found that leaf chlorophyll content decreases under heat stress, decreasing sucrose content [16]. The level of *chl* synthesis and breakdown under heat stress determines genotypes' tolerance levels [17]. HS varieties with damaged chlorophyll synthesis pathways lead to decreased chlorophyll levels. As shown in Table 5, the chlorophyll content of all varieties shows a downward trend after high-temperature treatment for 7 days. Numerous stresses often trigger the occurrence of photoinhibition in plants, further degrading chlorophyll and the photosynthetic system antenna proteins. As shown in Table 4, we found that Chlorophyll a (r = −0.836) and Chlorophyll b (r = −0.797) were negatively correlated with the heat damage index. The negative correlation between Chlorophyll a and the heat damage index was higher, indicating that Chlorophyll a content reflects plants' heat damage degree more accurately.

**Table 4.** Correlation analysis between various physiological indicators and heat damage index. * indicate significant difference ($p \leq 0.05$), and ** indicates a highly significant difference ($p \leq 0.01$).

| Correlation | Fresh Weight | Dry Weight | Fv/Fm | MDA | Ascorbic Acid | Chlb | Tr | Root Activity | Root Shoot Ratio | Heat Damage Index | Leaf Area | Plant Height | Plant Width | Ci | Gs | Pn | Chla | SS | VC | SP |
|---|---|---|---|---|---|---|---|---|---|---|---|---|---|---|---|---|---|---|---|---|
| fresh weight | 1 | | | | | | | | | | | | | | | | | | | |
| dry weight | 0.730 ** | 1 | | | | | | | | | | | | | | | | | | |
| Fv/Fm | 0.611 ** | 0.510 * | 1 | | | | | | | | | | | | | | | | | |
| MDA | −0.580 ** | −0.485 * | −0.576 ** | 1 | | | | | | | | | | | | | | | | |
| ascorbic acid | 0.813 ** | 0.650 ** | 0.599 ** | −0.657 ** | 1 | | | | | | | | | | | | | | | |
| chlb | 0.711 ** | 0.567 ** | 0.638 ** | −0.735 ** | 0.826 ** | 1 | | | | | | | | | | | | | | |
| Tr | 0.734 ** | 0.580 ** | 0.693 ** | −0.778 ** | 0.675 ** | 0.622 ** | 1 | | | | | | | | | | | | | |
| root activity | 0.595 ** | 0.680 ** | 0.531 * | −0.841 ** | 0.741 ** | 0.735 ** | 0.695 ** | 1 | | | | | | | | | | | | |
| root shoot ratio | 0.489 * | 0.559 * | 0.682 ** | −0.39 | 0.603 ** | 0.719 ** | 0.402 | 0.536 * | 1 | | | | | | | | | | | |
| heat damage index | −0.799 ** | −0.766 ** | −0.735 ** | 0.628 ** | −0.859 ** | −0.797 ** | −0.679 ** | −0.684 ** | −0.631 ** | 1 | | | | | | | | | | |
| leaf area | 0.550 * | 0.552 * | 0.583 ** | −0.693 ** | 0.678 ** | 0.704 ** | 0.550 * | 0.763 ** | 0.606 ** | −0.619 ** | 1 | | | | | | | | | |
| Plant hight | 0.594 ** | 0.425 | 0.708 ** | −0.607 ** | 0.665 ** | 0.696 ** | 0.595 ** | 0.678 ** | 0.683 ** | −0.604 ** | 0.727 ** | 1 | | | | | | | | |
| plant width | 0.305 | 0.449 * | 0.367 | −0.514 * | 0.3 | 0.491 * | 0.371 | 0.505 * | 0.374 | −0.414 | 0.621 ** | 0.206 | 1 | | | | | | | |
| Ci | 0.611 ** | 0.586 ** | 0.681 ** | −0.402 | 0.677 ** | 0.676 ** | 0.641 ** | 0.450 * | 0.623 ** | −0.673 ** | 0.402 | 0.446 * | 0.423 | 1 | | | | | | |
| Gs | 0.679 ** | 0.446 * | 0.594 ** | −0.488 * | 0.696 ** | 0.710 ** | 0.607 ** | 0.507 * | 0.628 ** | −0.682 ** | 0.287 | 0.554 * | 0.157 | 0.582 ** | 1 | | | | | |
| Pn | 0.713 ** | 0.594 ** | 0.837 ** | −0.674 ** | 0.825 ** | 0.800 ** | 0.720 ** | 0.641 ** | 0.632 ** | −0.843 ** | 0.632 ** | 0.684 ** | 0.395 | 0.705 ** | 0.757 ** | 1 | | | | |

**Table 4.** *Cont.*

| Correlation | Fresh Weight | Dry Weight | Fv/Fm | MDA | Ascorbic Acid | Chlb | Tr | Root Activity | Root Shoot Ratio | Heat Damage Index | Leaf Area | Plant Height | Plant Width | Ci | Gs | Pn | Chla | SS | VC | SP |
|---|---|---|---|---|---|---|---|---|---|---|---|---|---|---|---|---|---|---|---|---|
| Chla | 0.765 ** | 0.684 ** | 0.772 ** | −0.726 ** | 0.865 ** | 0.829 ** | 0.750 ** | 0.791 ** | 0.619 ** | −0.836 ** | 0.676 ** | 0.684 ** | 0.472 * | 0.737 ** | 0.720 ** | 0.931 ** | 1 | | | |
| SS | 0.708 ** | 0.641 ** | 0.864 ** | −0.665 ** | 0.773 ** | 0.815 ** | 0.675 ** | 0.681 ** | 0.689 ** | −0.855 ** | 0.697 ** | 0.698 ** | 0.498 * | 0.659 ** | 0.673 ** | 0.924 ** | 0.926 ** | 1 | | |
| VC | 0.813 ** | 0.650 ** | 0.599 ** | −0.657 ** | 1.000 ** | 0.826 ** | 0.675 ** | 0.741 ** | 0.603 ** | −0.859 ** | 0.678 ** | 0.665 ** | 0.3 | 0.677 ** | 0.696 ** | 0.825 ** | 0.865 ** | 0.773 ** | 1 | |
| SP | 0.610 ** | 0.656 ** | 0.885 ** | −0.622 ** | 0.577 ** | 0.684 ** | 0.707 ** | 0.679 ** | 0.691 ** | −0.695 ** | 0.631 ** | 0.730 ** | 0.452 * | 0.645 ** | 0.619 ** | 0.835 ** | 0.847 ** | 0.896 ** | 0.577 ** | 1 |

**Table 5.** The content of MDA, chlorophyll a, chlorophyll b, ascorbic acid of *Brassica rapa* cultivars with elevated heat tolerance under heat stress conditions. Different letters in a–e represent significant differences ($p < 0.05$) using one-way ANOVA. Comparisons between the mean values were made by the least significant difference.

| Variety Name | MDA | Chlb | Chla | Ascorbic Acid |
|---|---|---|---|---|
| shanghaiqing | 5.50 ± 0.07 a | 0.18 ± 0.00 bcd | 0.49 ± 0.02 bcde | 0.22 ± 0.01 bcde |
| wuyueman | 5.49 ± 0.04 a | 0.17 ± 0.08 bcd | 0.46 ± 0.02 bcde | 0.20 ± 0.00 bcde |
| aijiaohuang | 5.73 ± 0.14 a | 0.19 ± 0.01 bcd | 0.42 ± 0.01 bcde | 0.19 ± 0.00 bcde |
| baimeigui | 4.49 ± 0.06 b | 0.24 ± 0.01 bc | 0.53 ± 0.01 bcde | 0.25 ± 0.01 bcde |
| dongguanmei | 4.62 ± 0.05 b | 0.26 ± 0.01 b | 0.62 ± 0.01 bcde | 0.31 ± 0.00 bc |
| gaohuaqing | 2.85 ± 0.33 bcd | 0.37 ± 0.01 a | 0.82 ± 0.01 bc | 0.34 ± 0.01 b |
| jiaoyang | 2.98 ± 0.09 bcd | 0.39 ± 0.01 a | 0.87 ± 0.01 b | 0.37 ± 0.00 a |
| xinxiaqing | 3.62 ± 0.05 bc | 0.36 ± 0.00 a | 0.83 ± 0.01 bc | 0.36 ± 0.00 bc |
| Kangre605 | 3.51 ± 0.00 bc | 0.32 ± 0.01 a | 0.72 ± 0.01 bcd | 0.34 ± 0.00 bc |
| jinfei | 5.01 ± 0.00 b | 0.28 ± 0.00 bc | 0.55 ± 0.01 bc | 0.21 ± 0.02 bcde |
| liehuojingang | 3.36 ± 2.89 bc | 0.38 ± 0.10 a | 1.04 ± 0.01 a | 0.38 ± 0.00 a |
| dongxing | 3.84 ± 0.33 bc | 0.29 ± 0.00 bc | 0.76 ± 0.02 bcd | 0.24 ± 0.00 bcde |
| suzhouqing | 4.87 ± 0.14 b | 0.31 ± 0.01 b | 0.46 ± 0.01 bcde | 0.24 ± 0.01 bcde |
| jinbao | 4.64 ± 0.05 b | 0.25 ± 0.01 bc | 0.60 ± 0.02 bcde | 0.30 ± 0.00 bc |
| chibai | 5.07 ± 0.05 b | 0.34 ± 0.01 b | 0.65 ± 0.02 bcde | 0.31 ± 0.00 bc |
| siyueman | 4.05 ± 0.06 bc | 0.28 ± 0.01 bc | 0.47 ± 0.01 bcde | 0.24 ± 0.00 bcde |
| xinlvxiu | 3.50 ± 0.01 bc | 0.22 ± 0.01 bc | 0.46 ± 0.02 bcde | 0.20 ± 0.00 bcde |
| xiali | 4.30 ± 0.07 bc | 0.28 ± 0.01 bc | 0.59 ± 0.00 bcde | 0.29 ± 0.00 bcde |
| aoxia | 4.32 ± 0.10 bc | 0.27 ± 0.01 bc | 0.63 ± 0.01 bcde | 0.31 ± 0.00 bc |
| hanxiu | 5.14 ± 0.11 a | 0.25 ± 0.01 bc | 0.55 ± 0.02 bcde | 0.27 ± 0.01 bcde |

We also determined the SS and SP contents of different *Brassica rapa* materials under heat stress. Table 4 shows that the SS contents of four varieties of *Aoxia*, *Xinxiaqing*, *jiaoyang*, and *liehuojingang* were significantly higher. The SS contents of HS varieties *xinlvxiu*, *aijiaohuang*, and *wuyueman* were lower than other varieties. Studies have reported that SS accumulation in cells can greatly enhance plants' stress tolerance [16]. Furthermore, higher SS content plays an important role in maintaining proline content and ROS catabolism balance.

Higher concentrations of SS in HR plants are conducive to maintaining cell homeostasis and normal metabolism. In this study, differences in SP content between different varieties are more apparent. SS and SP are deemed to function as osmotic regulatory substances in vivo, protecting the integrity of the cell membrane. Therefore, higher SS and SP levels help protect cell membrane systems and enhance heat tolerance. They maintain osmoregulation and avoid damage to protein structures within the cell. Our data showed that the heat damage index was highly negatively correlated with SS (r = −0.86) and SP (r = −0.695) (Table 4) contents. In addition, a significant negative correlation was found between ascorbic acid and plants' heat damage index, indicating that heat-resistant varieties have higher levels of ascorbic acid (r = −0.859) and stronger antioxidant abilities (Table 4). SS and ascorbic acid levels have been proposed as important indicators for identifying the degree of heat damage.

Free radicals in plants cause membrane lipid peroxidation and produce malondialdehyde when exposed to high temperatures. Heat stress often leads to membrane lipid peroxidation and increased MDA levels [18].

As shown in Table 3, MDA content significantly accumulated in HS plants (*aijiaohuang*, *xinlvxiu*, and *wuyueman*). On the contrary, HR plants (*jiaoyang*, *liehuojingang*, *aoxia*, and *xinxiaqing*) showed low MDA levels. This difference indicates that the stability of HSA plants' plasma membrane is damaged and more sensitive to heat stress. In this study, HS plants had higher REC levels. The correlation between MDA (r = 0.628) and the heat damage index is not as strong as that for other indicators (Table 4).

*3.4. Gas-Exchange Parameters of Different B. rapa Cultivars under Heat Stress*

Heat stress destroys plants' photosynthetic system, reducing their photosynthetic rate and stomatal opening and increasing their transpiration rate. In this study, twenty *B. rapa* cultivars were exposed to high-temperature treatment for 7 days. Gas exchange indexes were measured, including photosynthetic rate (Pn), transpiration rate (Tr), and intercellular $CO_2$ concentration (Gs). HR plants' photosynthetic rate was significantly higher than HS plants. Among them, *liehuojingang* had the highest Pn (18.76 µmol m$^{-2}$ s$^{-1}$), followed by *jiaoyang* (16.9279 µmol m$^{-2}$ s$^{-1}$) and *xinxiaqing* (16.1187 µmol m$^{-2}$ s$^{-1}$). HS plants' Pn (*xinluxiu*, *wuyueman*, *aijiaohuang*, and *shanghaiqing*) was approximately 10.5 µmol m$^{-2}$ s$^{-1}$. Many studies have revealed that heat stress enhances the transpiration rate of plants, leading to death by excessive dehydration [19]. We measured the E levels of these *B. rapa* cultivars under heat stress. As shown in Table 5, the E levels of *liehuojingang*, *jiaoyang*, *xinxiaqing*, and *gaohuaqing* were relatively higher than those of other varieties. The E levels of *liehuojingang* (4.35 gm$^2$-h) were almost twice that of *xinlvxiu* (2.24 m$^2$-h). HR plants' Gs levels were significantly higher than HS varieties. Stomata are the main channels for the gas exchange of plant leaves. HR plants have a significantly higher level of Ci than HS plants. These results suggest that HR plants under heat stress have a stronger capacity for photosynthetic gas exchange than HS plants. In summary, Pn (r = −0.843) is more suitable for plant heat tolerance identification than E (r = −0.679), Ci (r = −0.682), or Gs (r = −0.673).

Heat stress considerably harms the quantum yield of PS(II) photochemistry. The Fv/Fm value of plants can be used as an important indicator for identifying the degree of multiple abiotic stresses. After heat stress treatment, we measured the Fv/Fm values of these 20 *Brassica rapa* varieties. Higher Fv/Fm values in HR plants represented higher PSII efficacy relative to HS plants. The Fv/Fm of *jiaoyang* (0.33) was twice as much as *aijiaohuang* (0.78). Meanwhile, Fv/Fm was highly correlated with the heat damage index (r = −0.735).

We conducted a cluster analysis on all *Brassica rapa* based on these physiological indicators (Figure 2). We found that the heat resistance of *xinxiaqing* and *gaohuaqing*, as well as *jiaoyang* and *liehuojingang*, was relatively consistent, whereas heat sensitivity was relatively consistent for *wuyueman*, *aijiaohuang*, and *xinlvxiu*. These materials were divided into seven heat-resistant varieties and thirteen heat-sensitive varieties. We also analyzed the correlation between the heat damage index and physiological growth indicators of *Brassica rapa* under heat stress (Table 6). The correlation analysis showed that SS, ascorbic acid, photosynthetic rate, dry and fresh weight, and Fv/Fm are useful heat tolerance indicators (Table 6).

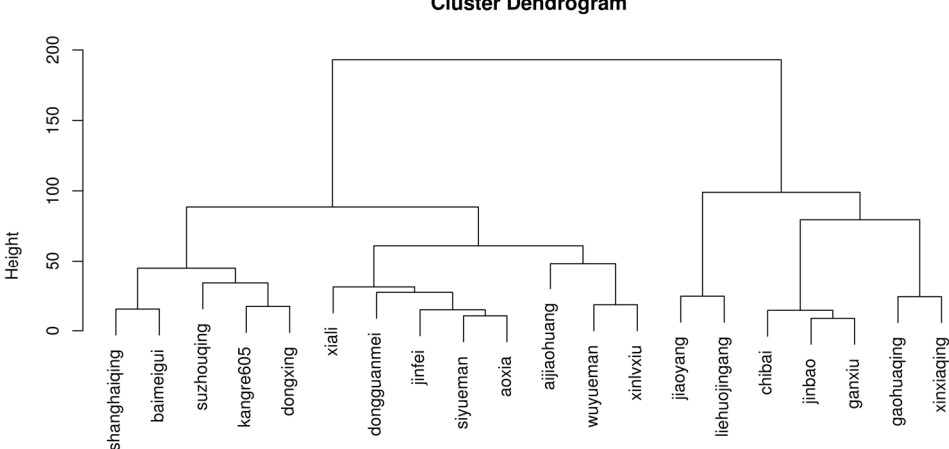

**Figure 2.** Cluster analysis of *Brassica rapa* varieties based on heat tolerance index.

**Table 6.** Photosynthetic rate (A), transpiration rate (E), Intercellular $CO_2$ concentration (Ci), stomatal conductance (Gs) and Pearson coefficient (r) with heat damage index of *Brassica rapa* seedlings under heat stress, the data represent mean values ±SE from four independent experiments. Different letters in a–e represent significant differences ($p < 0.05$) using one-way ANOVA. Comparisons between the mean values were made by the least significant difference.

| Variety Name | Pn | Tr | Gs | Ci |
| --- | --- | --- | --- | --- |
| shanghaiqing | 11.57 ± 4.02 bc | 2.72 ± 0.06 b | 254.85 ± 1.51 bcde | 374.67 ± 1.75 bc |
| wuyueman | 9.94 ± 0.53 bc | 2.57 ± 0.07 b | 241.00 ± 3.26 bcde | 308.14 ± 1.45 bcde |
| aijiaohuang | 10.44 ± 0.40 bc | 2.79 ± 0.077 b | 281.04 ± 6.86 bcde | 334.81 ± 2.08 bcde |
| baimeigui | 11.33 ± 0.27 bc | 3.76 ± 0.06 b | 245.27 ± 4.01 bcde | 369.811 ± 0.87 bcd |
| dongguanmei | 13.42 ± 0.42 bc | 2.91 ± 0.01 bc | 249.82 ± 3.71 bcde | 357.97 ± 1.58 bcde |
| gaohuaqing | 16.93 ± 0.78 b | 3.68 ± 0.08 bc | 395.21 ± 3.99 a | 365.38 ± 0.73 bcde |
| jiaoyang | 15.36 ± 0.43 b | 4.15 ± 0.04 bcd | 344.28 ± 3.01 bcde | 439.41 ± 0.43 bc |
| xinxiaqing | 16.12 ± 0.62 b | 3.96 ± 0.11 bc | 372.86 ± 2.82 b | 361.53 ± 1.01 bcde |
| Kangre605 | 14.61 ± 0.17 b | 3.74 ± 5.60 a | 270.18 ± 3.52 bcde | 369.81 ± 0.87 bcd |
| jinfei | 12.15 ± 0.53 bc | 2.62 ± 0.10 bc | 237.26 ± 5.94 bcde | 336.09 ± 2.22 bcde |
| liehuojingang | 18.76 ± 030 ac | 4.35 ± 0.03 bc | 365.24 ± 4.79 bc | 449.77 ± 5.15 a |
| dongxing | 15.60 ± 0.26 b | 3.32 ± 0.13 bc | 281.91 ± 5.27 bcde | 376.78 ± 1.50 b |
| suzhouqing | 12.750 ± 0.25 bc | 2.91 ± 0.03 bc | 284.66 ± 3.86 bcde | 384.44 ± 1.60 b |
| jinbao | 13.87 ± 0.29 bc | 3.37 ± 0.05 b | 328.47 ± 4.84 bcde | 368.29 ± 3.561 bcd |
| chibai | 12.504 ± 0.40 bc | 2.46 ± 0.02 bc | 331.84 ± 5.98 bcd | 375.511 ± 1.65 bc |
| siyueman | 10.47 ± 0.39 bc | 3.05 ± 0.03 b | 246.4 ± 3.52 bcde | 338.16 ± 1.98 bcde |
| xinlvxiu | 10.57 ± 0.31 b | 3.25 ± 0.01 b | 254.99 ± 3.94 bcde | 306.54 ± 2.08 bcde |
| xiali | 14.28 ± 0.24 bc | 3.03 ± 0.03 b | 266.68 ± 4.16 bcde | 337.47 ± 4.38 bcde |
| aoxia | 12.75 ± 0.25 bc | 2.76 ± 0.02 bc | 244.54 ± 3.59 bcde | 346.96 ± 2.65 bcde |
| hanxiu | 13.51 ± 0.28 bc | 3.29 ± 0.17 b | 323.42 ± 3.67 bcde | 365.45 ± 2.99 bcde |

*3.5. Biochemical Analysis of Enzymatic Activities and Hydroxyl Scavenging Capacity*

Upon exposure to high-temperature environments, plants perceive heat stress signals and trigger a series of transcriptional responses. It is widely accepted that heat stress disrupts ROS metabolic balance, resulting in excessive accumulation and instability in the membrane system. Plants initiate antioxidant mechanisms that help counteract the toxic effects of ROS [20]. Plants produce a range of antioxidants to scavenge ROS and protect cells from oxidative stress. The production of antioxidant enzymes, such as superoxide catalase (CAT), dismutase (SOD), or peroxidase (POD), is essential for eliminating super-oxide radicals. In this study, we mainly studied the differences between antioxidant and photosynthetic systems in heat-resistant and heat-sensitive varieties under heat stress. We measured peroxidase (POD), catalase (CAT), ascorbic acid oxidase (AAO), and superoxide dismutase (SOD) activity, proline (Pro) and hydrogen peroxide ($H_2O_2$) content, as well as the scavenging rate of hydroxyl free radicals in HR and HS plants under heat stress. As shown in Figure 3, CAT and SOD enzyme activities tend to increase gradually, whereas POD and AAO enzyme activities increase first and then decrease. This result indicates that POD and AAO enzyme activities are temporarily maintained in plants' heat stress response. CAT and SOD enzymes play a leading role in continuous high-temperature environments. Our results showed that HR varieties' protective enzyme activity was higher, preventing cell damage by scavenging ROS. Previous findings showed that the expression of ROS scavenging genes is strongly upregulated in heat-tolerant plants, which was in accordance with our data [21]. In addition, we found that the hydroxyl radical scavenging rate of HR varieties was higher than that of HS varieties. The $H_2O_2$ content of HS varieties was significantly higher than in HR varieties, which aligns with previous enzyme activity assay results (Figure 4). To our knowledge, higher $H_2O_2$ content can destroy cellular structures and produce active oxygen radicals that damage cells [22].

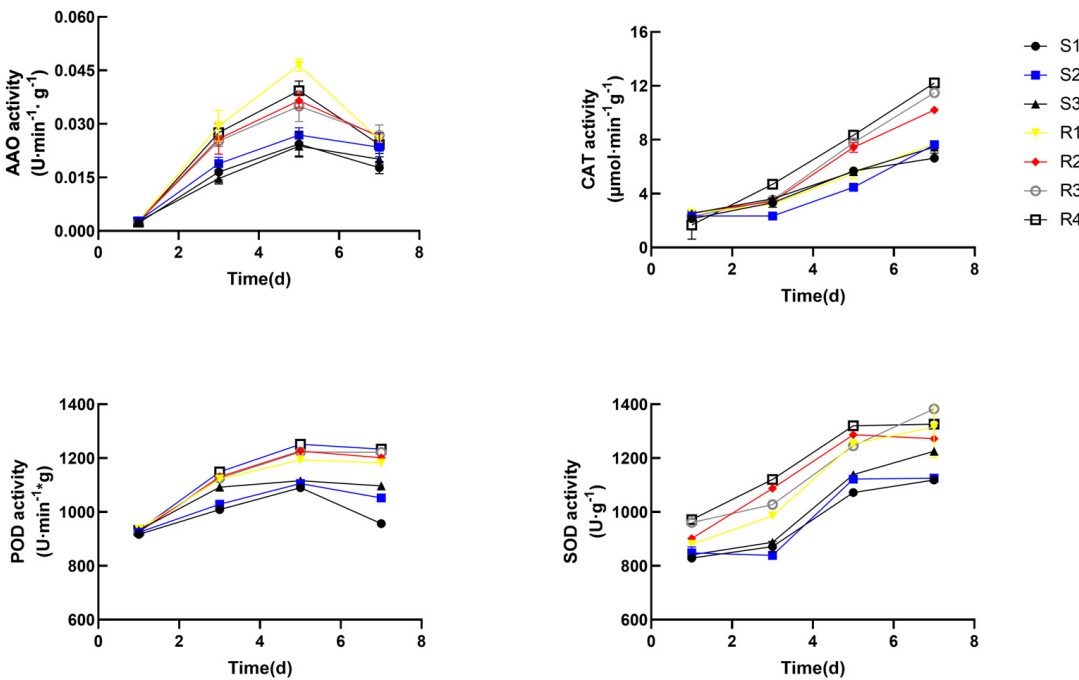

**Figure 3.** The antioxidant enzyme activity of ascorbic acid oxidase (AAO), catalase (CAT), Peroxidase (POD) and superoxide dismutase (SOD) of *Brassica rapa*. cultivars under heat stress.

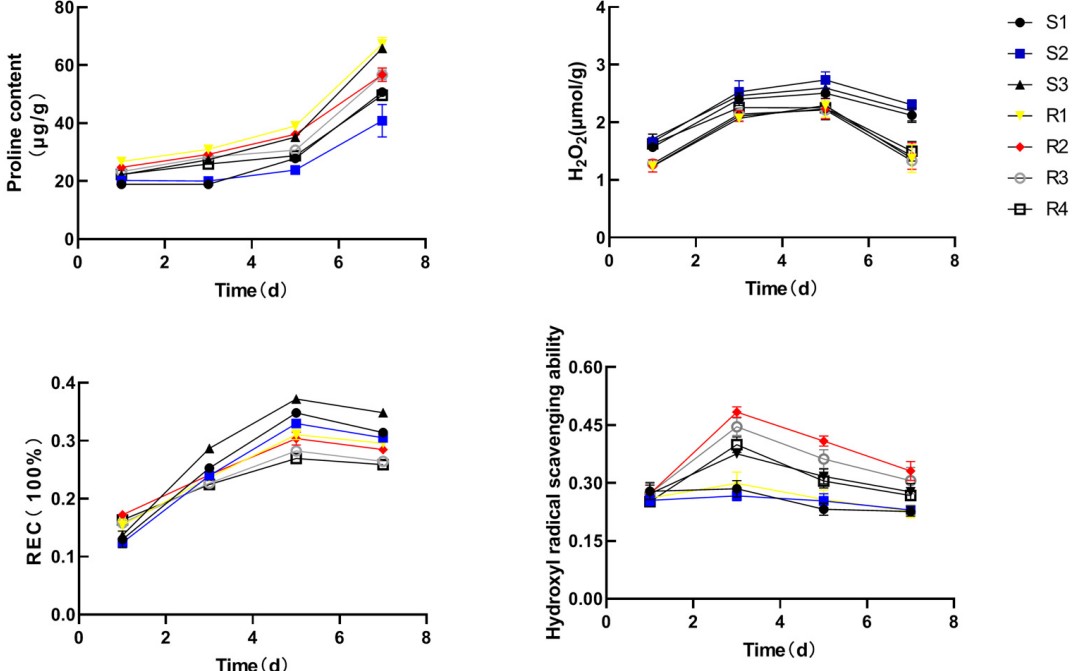

**Figure 4.** Proline content, hydrogen peroxide $H_2O_2$ content, relative conductivity and hydroxyl radical scavenging ability of *Brassica rapa*. cultivars under heat stress.

Proline is an important amino acid that helps protect the cell membrane structure and participates in antioxidant pathways [23,24]. Under heat stress, Pro accumulation gradually increased and rose quickly after 3 days of treatment. Under heat stress, Pro adjusted the osmotic pressure balance in vivo. To understand the differences in cell membrane permeability, we measured the relative conductivity of heat-resistant and heat-sensitive varieties' leaves under heat stress. Our results showed that the relative conductivity of HR and HS varieties' leaves increased continuously for the first 5 days, then gradually decreased.

Moreover, the relative conductivity of heat-sensitive varieties' leaves is higher than that of heat-resistant varieties, indicating that HS varieties have high membrane permeability under heat stress. Therefore, CAT, CAT activity, the hydroxyl radical scavenging rate, and pro and $H_2O_2$ contents were suitable for identifying heat tolerance in *Brassica rapa* varieties. ROS accumulation disrupts photosynthesis and respiration and deactivates biological macromolecule activity [25]. These results suggest that HR plants possess superior antioxidant systems and a stronger tolerance than HS plants.

### 3.6. The PSII and PSI Activity of Brassica rapa Leaves under Heat Stress

Chlorophyll fluorescence parameters have been widely used in photosynthesis research. Chlorophyll fluorescence parameters can assess the primary reaction processes of photosynthesis and electron transfer. Almost all photosynthetic changes are reflected in chlorophyll fluorescence. We measured the quantum yield [Y(I), Y(II)] and electron transport rate [ETR(I), ETR(II)] of PSI and PSII in these *Brassica rapa* materials, respectively. Multiple abiotic stresses, including heat, drought, cold, salinity, and heavy metals can affect the photosynthetic electron transport process. These stresses can reduce the efficiency of the photochemical reaction and produce excess absorbed light energy, aggravating photoinhibition [19,26–28]. We also measured the rapid light curves (RLCs) of photosynthetic quantum yields. Heat stress has a destructive impact on photosystems, especially PSII, and causes photoinhibition. As shown in Figure 5, ETR (II) and ETR (I) were significantly lower in HS (S1, S2, S3) plants than in HR (R1, R2, R3, R4) plants. Our results showed that photosynthetic electron transport activity varied between HS and HR varieties. HR plants can maintain higher photosynthetic efficiency and biomass accumulation than HS plants. PSII and PSI activities from *Brassica rapa* leaves in HR plants were higher than in HS plants. The effective quantum yield of PSII photochemistry [Y(II)] increased initially and then decreased gradually with increasing light intensity. By contrast, Y(II) was significantly lower in the HS group than in the HR group at almost all light intensities.

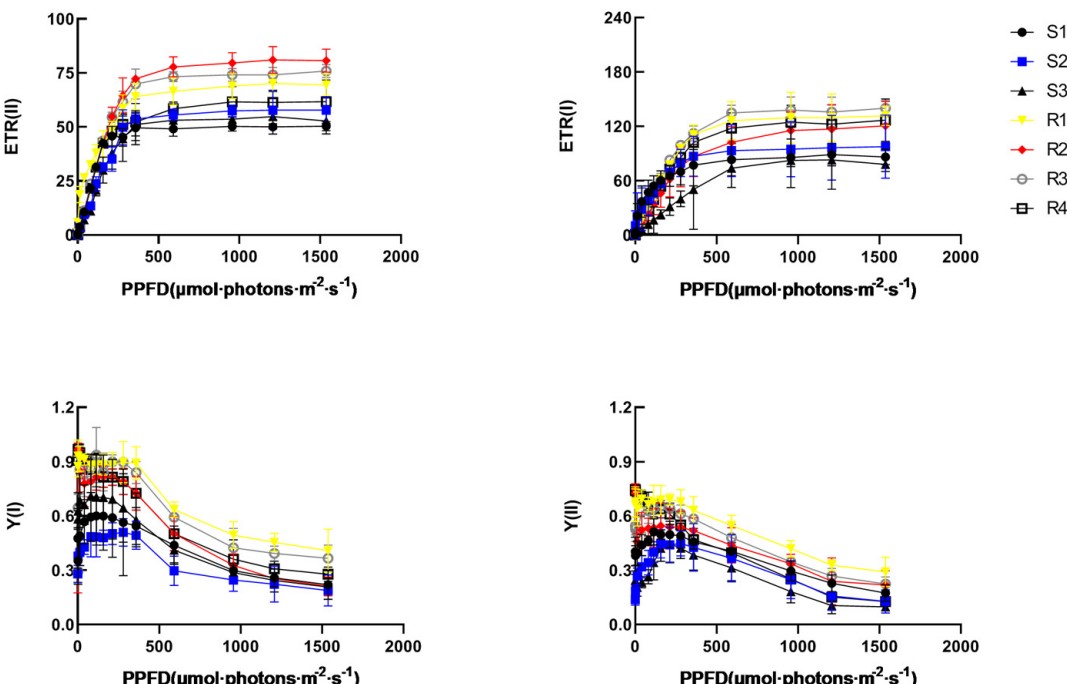

**Figure 5.** Energy distribution in photosystems electron transport rate of *Brassica rapa* under heat stress. Y(II), efficient quantum yield of PSII;Y(I), quantum yield of PSI; ETR(II), electron transport rate of PSII; ETR(I), electron transport rate of PSI.

As shown in Figure 6, the donor-side limitation [Y(ND)] gradually increased with increasing light intensity. The acceptor-side limitation [Y(NA)] exhibits an opposite trend to Y(ND). These results indicate that HR plants have a higher quantum yield of PSI non-photochemical energy dissipation. HR plants also have a stronger ability to induce PSII's non-photochemical quenching and activate PSI's self-protection mechanism. Plants' energy utilization mechanism was disturbed under heat stress, resulting in photoinhibitory effects.

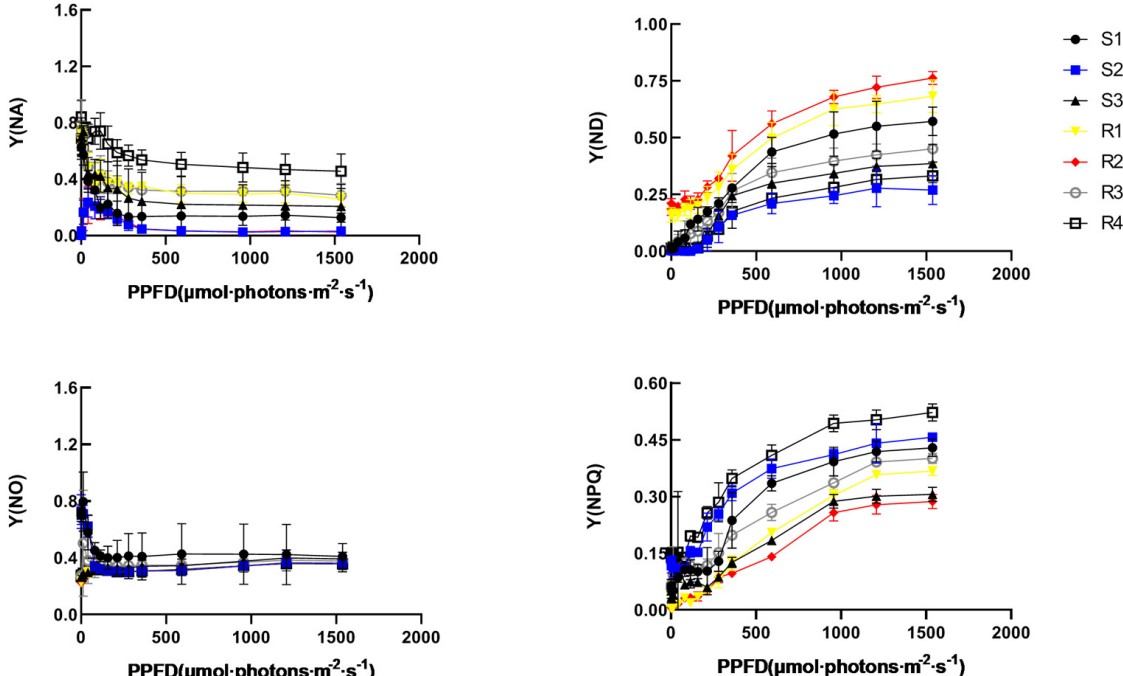

**Figure 6.** SI;PSINon−photochemical quenching (NPQ), Non−photoprotective heat dissipation Y(NO), the acceptor-side limitation [Y(NA)], donor side limitation of PSI Y(ND) from light curve analysis by dual−PAM of two *Brassica rapa* cultivars under heat stress. Data are the means of four replicates with standard errors shown by vertical bars.

The quantum yield of non-regulated energy dissipation in PSII [Y(NO)] was decreased and then kept stable with increasing light intensity. R1 and R2 were significantly higher than other groups before light intensity at 1000 μmol m$^{-2}$ s$^{-1}$. With increasing light intensity, the quantum yield of regulated energy dissipation in PSII [Y(NPQ)] increased rapidly. The Y(NPQ) of *jiaoyang* and *xinxiaqing* acquired a maximum value. Our results suggest that HR plants have a stronger capacity to induce PSII photoprotection. The quantum yield of PSI photochemistry [Y(I)] gradually decreased with increasing light intensity. Compared to HR plants, functional PQ pools were significantly reduced in HS plants.

## 4. Discussion

Increased population growth and atmospheric pollution have contributed to global warming, which has become a major limiting factor in crop production [29–31].

High-throughput sequencing technology has rapidly developed in recent decades. RNA sequencing and transgenic technology have been powerful tools for exploring potential molecular regulation mechanisms in abiotic stress [32–35]. However, using biotechnologies to develop new species and applying them to crop production is challenging. Therefore, it is essential to elucidate the heat tolerance mechanism in *Brassica rapa* from the viewpoint of physiology and biochemistry to develop new varieties.

Studying plants' heat-resistant mechanisms at the physiological and biochemical levels alongside molecular biology is important. On the other hand, establishing simple and effective variety identification methods is also crucial.

In the present study, heat-sensitive (HS) and heat-resistant (HR) plants were exposed to high-temperature environments (37 °C) for 7 days. We determined that morphological indicators such as plant height, width, leaf area, and dry and fresh weight represented the biomass of these materials under heat stress. We also measured the plants' root activity, and shoot root ratio to compare the effects caused by heat stress on these varieties' root growth. The chlorophyll content, soluble sugar (SS), proline, soluble protein (SP), ascorbic acid, and malondialdehyde (MDA) was measured. In addition, gas exchange parameters, including photosynthesis, transpiration rate (E), intercellular carbon dioxide concentration (Ci), and stomatal conductance (Gs) were measured to understand photosynthetic attributes' contribution to heat stress tolerance in *Brassica rapa*. Combined analyses of morphological and physiological indicators were performed to establish a simple and effective identification method. We measured their heat damage index and determined the relationship between *chl* content, MDA content, SS and SP content, growth attributes (including plant height, width, leaf area, dry and fresh weight, root-shoot ratio, root activity, etc.), gas exchange parameters, and heat damage indexes. These analyses helped us understand the effects of photosynthetic electron transfer and antioxidant pathways on plant heat tolerance. We selected several typical heat-sensitive and heat-resistant varieties and compared their differences in antioxidant enzyme activity, including AAO, CAT, SOD, and POD. The electron transfer rate and efficiency of the photosynthetic system were also surveyed.

Based on our data, we found that the growth and development of HS varieties were more negatively impacted by heat damage. Secondly, the *chl* content of HS was lower than in HR plants, indicating that *chl* degradation activity was more frequent in HS plants. Chlorophyll content directly influenced the plants' photosynthesis. The biomass of HR plants was greater than that of HS, including dry and fresh weight, plant height, plant width, leaf area, and root–shoot ratio. Our results suggest that the fixed carbon capacity is stronger in HR plants. Furthermore, there were significant differences in root development between heat-resistant and heat-sensitive varieties under heat stress. Roots are plants' vegetative organs for absorbing water and nutrients. Damaged roots will result in water deficiency, malnutrition, and death. In general, plants' leaf and root development affects the source–flow–sink relationship and yield. We also analyzed HR and HS varieties' antioxidant and photosynthetic systems to determine the differences between their photosynthetic electron transfer and oxidation resistance.

Numerous studies have focused on the heat stress mechanism of *Brassica rapa*. However, few studies have established a simple and effective method for identifying heat-resistant varieties. Several studies have emphasized the molecular mechanism of heat tolerance, including the heat shock protein pathway, ROS pathway, hormone transduction pathway, etc. These results provide valuable information about the heat tolerance mechanism in plants. Based on these findings, we investigated the physiological and biochemical responses of *Brassica rapa* varieties with different heat tolerance levels to high-temperature stress. Our paper examines how plants adapt to heat stress by changing their osmoregulation substance catabolism and antioxidant and photosynthetic pathways. Our study provides a more accurate identification method for heat tolerance and emphasizes the importance of light protection and antioxidant mechanisms for plant heat tolerance. Our future research will focus on genes involved in related pathways.

Heat stress has devastating effects on cell activity [36,37]. However, specific physiological reactions that determine the heat tolerance mechanism have not been clarified. In this study, we compared the physiological and biochemical responses of HS and HR varieties to heat stress. Our results showed that protective enzymes and photosynthetic electron transport play important roles in the heat tolerance of *Brassica rapa*. The photosynthetic apparatuses of different *Brassica rapa* materials behave differently under heat stress. Therefore, the heat resistance between *Brassica rapa* materials also differs.

Our study provides valuable information on the physiobiochemical response of *Brassica rapa* to heat stress. By considering the gene regulation of antioxidant enzyme synthesis and photosynthetic electron transport pathways at the molecular level, future research

on the heat tolerance mechanism of *Brassica rapa* will provide an insight into the high-temperature adaptation of horticultural crops.

These findings also shed light on the physiological mechanisms of heat stress on *Brassica rapa* and may promote the genetic modification of heat tolerance in horticultural crops.

## 5. Conclusions

In summary, we recommend a method for screening heat-resistant varieties based on the correlation cluster analysis of certain physiological indicators (chlorophyll content, photosynthetic rate, soluble sugar and soluble protein content, dry and fresh weight, and Fv/Fm) and the heat damage index of *Brassica rapa*. Secondly, heat-resistant varieties have a stronger antioxidant system, which produces antioxidants to scavenge ROS in *Brassica rapa*. On the other hand, heat-resistant varieties of *B. rapa* acquire stronger heat tolerance by protecting the electron transport system in photosynthesis and enhancing antioxidant enzyme activity in ROS signaling. Our results provide a theoretical basis for the future development of heat-tolerant horticulture crop germplasm resources.

**Author Contributions:** J.Y. conducted the research and wrote the manuscript; Q.N. conceived the project; N.F., P.L. and S.T. performed data analysis and assisted in the experiments; L.C. performed microscopic studies. All authors have read and agreed to the published version of the manuscript.

**Funding:** This research and the APC was funded by [Shanghai Agriculture Applied Technology development program] grant number [G.20190202].

**Data Availability Statement:** Data sharing is not applicable to this article as no new data were created or analyzed in this study.

**Conflicts of Interest:** The authors declare no conflict of interest.

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
