# Peer review of "Identification of Heat-Resistant Varieties of Non-Headed Chinese Cabbage and Discovery of Heat-Resistant Physiological Mechanisms"

_horticulturae, doi:10.3390/horticulturae9060619_

Round 1

Reviewer 1 Report

1) Regarding this manuscript, the English and overall writing usage needs extensive revision. It seems the manuscript has not been checked carefully before submission. There are several examples of sentences and words with low-quality writing (regarding grammar or writing per se) and they should be carefully checked before the new submission:

Brassica rapa should be italicized (lines 11, 15, 21, and several others);

The leaf samples were collected at (line 92) - at what?

measurement  (line 107): this word should be should start with a capital letter;

Heat stress initiaHeat sHeat stress (line 217): I do not understand what this means at all

And several other examples

2) I recommend rewriting some long sentences. I think it would be better if these were short sentences. For example, there are sentences are too long like this one:

´This paper provides a more accurate identification method for heat tolerance, and explains the importance of light protection and antioxidant mechanism for plant heat tolerance, so as to provide the key genes for future exploration of related pathways` (lines 413 to 415).

3) In the Table 01,  regarding Serial number 1; Pn 11.57±4.02, the authors should provide the statistical letter.

4) In the Table caption, the authors should provide an explanation of the statistical letters and mention the statistical test that has been used.

5) Minor suggestion: regarding the manuscript title, maybe it would be better to use the word ´mechanisms` (plural). Either, this is up to the authors. 

 6) The authors mentioned: ´Under the heat stress treatment, leaf samples were collected at 1, 3, 5, 7 days after treatment`.  The explicit explanation for choosing such days for the analyses should be presented.

As said, the English and overall writing usage needs extensive revision.

Author Response

Dear Reviewers:

Thank you very much for your comments and suggestions. These suggestions are very valuable and helpful for revising and improving our paper, as well as the important guiding significance to our research. We have studied comments carefully and have made correlation which we hope meet with approval. Here, we would like to explain the changes briefly as follows:

Comment: Brassica rapa should be italicized (lines 11, 15, 21, and several others);

Response: we have changed the “Brassica rapa ” to “Brassica rapa ” in this paper;

Comment:The leaf samples were collected at (line 92) - at what?

Response: we are very sorry for incorrecting writing, it is rectified as “leaf samples were collected and frozen in liquid nitrogen, and stored in -80 ℃super cold refrigerator”.

Comment: measurement  (line 107): this word should be should start with a capital letter.

Response:we have changed the “measurement” to “Measurement”

Comment:Heat stress initiaHeat sHeat stress (line 217): I do not understand what this means at all

Response: we are very sorry for this mistake, we have deleted this sentence.

Comment: 1) I recommend rewriting some long sentences. I think it would be better if these were short sentences. For example, there are sentences are too long like this one:

Response:I'm sorry for the difficulty in reading the paper due to our poor writing skills. We have found an English native speaker with a research background to review our manuscript during revision.

Comment: 2) Example:´This paper provides a more accurate identification method for heat tolerance, and explains the importance of light protection and antioxidant mechanism for plant heat tolerance, so as to provide the key genes for future exploration of related pathways` (lines 413 to 415).

We have revised this sentence as”This paper provides a more accurate identification method for heat tolerance, and emphasize the importance of light protection and antioxidant mechanism for plant heat tolerance. We propose to focus future research on genes involved in related pathways.”

Comment: 3) In the Table 01,  regarding Serial number 1; Pn 11.57±4.02, the authors should provide the statistical letter.

Response: We apologize for the incomplete data caused by our negligence. We have added lowercase letters: Pn 11.57±4.02bc

Comment: 4) In the Table caption, the authors should provide an explanation of the statistical letters and mention the statistical test that has been used.

Response: We are sorry for providing complete information, we have added “different letters in a-e represent significant differences (P< 0.05) using one-way ANOVA. Comparisons between the mean values were made by the least significant difference.” into the table caption.

Comment: 5) Minor suggestion: regarding the manuscript title, maybe it would be better to use the word ´mechanisms` (plural). Either, this is up to the authors. 

Response: According with your advice, we changed the title as ”Identification of heat-resistant varieties of Non-headed Chinese cabbage and exploration of heat-resistant physiological mechanisms”.

Comment: 6) The authors mentioned: ´Under the heat stress treatment, leaf samples were collected at 1, 3, 5, 7 days after treatment`.  The explicit explanation for choosing such days for the analyses should be presented.

Response:Thanks for your advice. To make information more complete, we added contents as “On the first day of treatment, there was no significant difference between heat sensitive and heat-resistant varieties. On the third day, there was a significant difference in heat damage symptoms, and on the fifth day, the difference intensified. On the seventh day, most of the heat sensitive varieties withered, while most of the heat-resistant varieties could still survive. Therefore, we took samples on the first, third, fifth, and seventh days for the determination of physiological indicators.”

Comments: on the Quality of English Language. As said, the English and overall writing usage needs extensive revision.

Response: We have found an English native speaker with a research background to review our manuscript during revision. And if you think there is any problem, you can raise it at any time. We will look for professional organizations to improve the language.

Reviewer 2 Report

The authors, using the example of a cold-resistant type of cabbage (20 varieties), studied its thermotolerance at the physiological and biochemical level, since, according to the authors, the system for assessing the heat resistance of green cabbage has not yet been established. However, similar works with other agricultural crops are found.

In the Conclusion, the authors write that “…conclude that chlorophyll content, photosynthetic rate, the content of soluble sugar and soluble protein, the dry and fresh weight, the Fv/Fm can be used for rapid screening of heat-resistant varieties in brassica rapa. » However, this does not meet the purpose of the study - "...to utilize physiological and biochemical indicators to establish an effective method for identifying heat-resistant varieties, which provided some information to explore the roles of other physiological attributes in heat stress tolerance in Brassica rapa." In fact, the authors recommend several methods, rather than one method, as stated in the purpose of the work.

 L.29-30. The authors write "At the end of the 21st century, the melting of glaciers raised the sea level to 59 cm."

You must provide a link to the literature source.

L. 44-46. The authors write "Under high temperature condition, Brassica rapa often shows slow growth, yellow and rotten leaves, increased fiber content, and serious death."

You must provide a link to the literature source.

L. 46. It is hardly possible to write “serious death” like that.

L. 56. It is necessary to decipher "HS and HR" - this is the first mention.

L. 56-70. Usually this information is not included in the introduction.

L. 76. It is not indicated how many plants were used in each variant of the experiment.

L. 87. What is "plant root activity"? - no explanation. The method "The plant root activity was measured by Methylene blue." is not stated.

L. 88. The total chlorophyll content was measured using a previously described method [6]. It is necessary to provide the essence of the method or a brief description. Reference [6] lacks a description of the method.

L. 89. Replace "root area" with "root surface area".

L. 97-98. The determining method for the relative conductance (REC) used in this study was described by Zheng et al [9].

This is not a well-known method, so it should be explained in more detail.

L. 101. How many plants were used to measure photosynthesis parameters?

L. 103. The experiment was conducted according to the manufacturer's procedures [11]. Reference [11] is a scientific article, not "the manufacturer's procedures". In the article [11] there is no methodology for determining the transpiration rate and stomatal conductance.

L. 111. What does "actinic red light" mean? Actinic light is light from the sun. How did the authors get "actinic red light" of varying intensity?

L. 113. Full name "RLCs" required at first mention.

L. 113. The RLCs of photosynthetic parameters were measured……

The authors did not indicate the device with which the measurements were taken.

L. 121. It is necessary to decipher "ST and MT" - this is the first mention.

L. 123. It is necessary to indicate the analytical and biological replicates, as well as how many experiments were carried out.

L. 131. Photographs of control plants and plants (HR and HS) after heat stress (in stress dynamics) would have embellished the manuscript.

L. 142, 165. It is not clear from the Methods what "Heat damage index" and "root growth index" represent, as they were calculated.

L. 152. It is important for researchers to know the variety, not its number. Therefore, in tables 1, 3, 4, “serial number” should be replaced with the name of the variety.

L. 169. Need a link to a figure or table.

L. 178. «As shown in Table 3, HR plants under heat stress showed a higher chlorophyll level relative to HS cultivars with…..»

However, it is not clear from the table which varieties are HR and HS.

L. 181-182. «Previous study found that the HR varieties have a relatively higher chlorophyll content compared to HS varieties.»

It is not clear which previous studies are in question.

L. 188. Chlorophyll a (r=-0.836) and Chlorophyll b (r=-0.797) are negatively correlated with the heat damage index,…. Where does this data come from? No reference to figure or table.

L. 193. From the Table 4, the SS content in the four varieties of Aoxia, Xinxiaqing, jiaoyang and liehuojingang……

L. 193. «From the Table 4, the SS content in the four varieties of Aoxia, Xinxiaqing, jiaoyang and liehuojingang were significantly higher, the SS content of HS varieties xinlvxiu…».

Table 4 does not contain data on the content of soluble sugars and proteins in leaves cabbage.

L. 203. «The SP level of HR varieties is twice than that of HS varieties.»

Where is this data?

L. 208. «Our data showed that the heat damage index was highly negatively correlated with the content of SS (r=-0.86) and SP (r=-0.695).»

These data are missing from Table 5.

L. 217. «Heat stress initiaHeat sHeat stress initially….»

There is an error here.

L. 218-220. Sentences duplicate each other - repetition.

L. 230. «In this work, the high temperature treatment was implemented on twenty B.rapa cultivars for 24 h.»

However, the Methods indicate the dynamics - from 1 to 7 days. It should be specified in the Methods.

L. 223. «As shown in the Table 3, the MDA content was significantly accumulated in HS plants (aijiaohuang, xinlvxiu and wuyueman).»

This is not visible from the table, because cultivar names are not given.

The absence of variety names in tables 1, 3 and 4 makes it difficult to review.

L. 308. «4.1. Biochemical analysis of enzymatic activities and Hydroxyl scavenging capacity, et al.»

Why is there a Results section in the Discussion (Figures 2, 3, 4)?

Author Response

Dear Reviewers:

Thank you very much for your comments and suggestions. These suggestions are very valuable and helpful for revising and improving our paper, as well as the important guiding significance to our research. We have studied comments carefully and have made correlation which we hope meet with approval. Here, we would like to explain the changes briefly as follows:

Thank you for your careful review and suggestions, which have helped us improve the quality of our paper. Many researchers have established methods for identifying germplasm resources in many horticultural crops. Due to changes in the ecological environment, identifying planting resources is indeed meaningful for stress resistance breeding. We hope that our work can enhance some useful reference value for the identification of heat resistant varieties of cruciferous plants in the future.

Comment: In the Conclusion, the authors write that “…conclude that chlorophyll content, photosynthetic rate, the content of soluble sugar and soluble protein, the dry and fresh weight, the Fv/Fm can be used for rapid screening of heat-resistant varieties in brassica rapa. » However, this does not meet the purpose of the study - "...to utilize physiological and biochemical indicators to establish an effective method for identifying heat-resistant varieties, which provided some information to explore the roles of other physiological attributes in heat stress tolerance in Brassica rapa." In fact, the authors recommend several methods, rather than one method, as stated in the purpose of the work.

Response: Thank you for your suggestion. There is indeed a problem with the wording of this paragraph.

Comment: L.29-30. The authors write "At the end of the 21st century, the melting of glaciers raised the sea level to 59 cm."You must provide a link to the literature source.

Response: We are Sorry for our carelessness in not attaching the literature. We have already added the literature to the text.

Comment:It is hardly possible to write “serious death” like that.

Response: I'm sorry for the mistake here. I have corrected it as “Under high temperature condition, Brassica rapa often shows slow growth, yellow and rotten leaves, increased fiber content, and susceptible to viruses”.

Comment:L. 56. It is necessary to decipher "HS and HR" - this is the first mention.

Response I'm sorry for the negligence here.I have deciphered "HS and HR" as” heat sensitive and heat resistant”.

Comment:L. 56-70. Usually this information is not included in the introduction.

Response: Thank you for your useful suggestions. I have included this part in the discussion module.

Comment: L. 76. It is not indicated how many plants were used in each variant of the experiment.

Response: Sorry, this information is not complete enough. I have added “fifty seedlings with uniform growth were selected for each variety for heat treatment.” in the Materials and methods.

Comment:L. 87. What is "plant root activity"? - no explanation. The method "The plant root activity was measured by Methylene blue." is not stated.

Response: I'm sorry for the unclear explanation of this part. I have added “The plant root dehydrogenase activity is indicative of assess plant root activity.” in the result. I have added “ the plant root dehydrogenase activity, the content of hydrogen, proline and hydroxyl radical scavenging rate were measured by using assay kits (Comin, Suzhou, China)” in the Materials and methods.

Comment: L. 88. The total chlorophyll content was measured using a previously described method [6]. It is necessary to provide the essence of the method or a brief description. Reference [6] lacks a description of the method.

Response:We apologize for the unclear measurement method, We have improved the content “Chlorophyll was extracted in the dark using 10 ml ethanol (95%) at 4 C using 0.2g fresh leaf sample. The chlorophyll content was calculated using the absorbance (A) at 645 and 663 nm using 95% ethanol as the control. “

Comment:L. 89. Replace "root area" with "root surface area".

Response: Thank you for your suggestion. We find it very reasonable. We have replace "root area" with "root surface area" in the text.

Comment: L. 97-98. The determining method for the relative conductance (REC) used in this study was described by Zheng et al [9].

This is not a well-known method, so it should be explained in more detail.

Response: Thank you for your suggestion. We have provided additional details, as “Rinse Brassica rapa leaves three times with distilled water, then absorb surface moisture, cut long strips, avoid leaf veins, take 0.1g and soak in a tube containing 10ml of distilled water for 12 hours. Measure the conductivity of the solution ( R1 ).Then boil the tube in water for 30 minutes, cool to room temperature, and then measure the conductivity(R2). REC=R1/R2×100%”.

Comment: L. 101. How many plants were used to measure photosynthesis parameters?

Response:Thank you for your careful inspection. We have added “ The fourth full-developed leaf from four plants per variety was used in this measurement.” in the text.

Comment: L. 103. The experiment was conducted according to the manufacturer's procedures [11]. Reference [11] is a scientific article, not "the manufacturer's procedures". In the article [11] there is no methodology for determining the transpiration rate and stomatal conductance.

Response: We are sorry for the mistake caused by our carelessness. We revised the sentence as “The experiment was conducted according to the manufacturer's procedures with minor modifications as described by zhang et al. [12].”the transpiration rate and stomatal conductance was measured according to the manufacturer's procedures.

Comment: L. 111. What does "actinic red light" mean? Actinic light is light from the sun. How did the authors get "actinic red light" of varying intensity?

Response:Thank you very much for your careful review. We have made a mistake and have changed it to “. Different photosynthetic photon flux density (PPFD) were set in this experiment.”

Comment: L. 113. Full name "RLCs" required at first mention.

Response:We are sorry for not decipher it here, we have added the full name” rapid light curves” in the text.

Comment:L. 113. The RLCs of photosynthetic parameters were measured……

The authors did not indicate the device with which the measurements were taken.

Response: We are sorry for not providing the information clearly. We have added “the rapid light curves (RLCs) of photosynthetic parameters were measured by DUAL-PAM-100 measuring systems, “ in the text.

Comment:L. 121. It is necessary to decipher "ST and MT" - this is the first mention.

Response: Thank you for your reminder. We have added “ single turnover flashes (ST, 50ms, PQ pools being oxidized) and multiple turnover flashes (MT, 50 ms, PQ pools are fully reduced) “ in the text.

Comment:L. 123. It is necessary to indicate the analytical and biological replicates, as well as how many experiments were carried out.

Response:We are sorry for not providing more detailed information. We have added “Quantitative assessment was conducted on randomly selected samples from four independent biological replicates. ” in the text.

Comment:L. 131. Photographs of control plants and plants (HR and HS) after heat stress (in stress dynamics) would have embellished the manuscript.

Response: Thank you for your thoughtful consideration. We have added the Photographs of control plants and plants (HR and HS) after heat stress.

Comment: L. 142, 165. It is not clear from the Methods what "Heat damage index" and "root growth index" represent, as they were calculated.

Response: Thanks for your kindly advice, we think it's very useful. We have added “According to plant performance, the heat damage index (HDI) is divided into five categories: Grade 0: the leaves have no obvious heat damage symptoms; Grade 1: The number of affected leaves in the plant is less than 1/3 of the total number of leaves in the plant; Grade 3: The number of affected leaves in the plant is greater than 1/3 of the total number of leaves and less than 1/2 of the total number of leaves in the plant; Grade 5: The number of affected leaves in the plant is greater than 1/2 of the total number of leaves and less than 2/3 of the total number of leaves in the plant; Grade 7: The number of damaged leaves in the plant is greater than 2/3 of the total number of leaves in the plant; HDI=∑(Number of plants per grade × grade)/ (Highest grade×total number of plants) × 100%;” in the metirials and methods. Additionally, “the shoot root ratio was calculated by the ratio of fresh or dry weight between the underground and aboveground parts of a plant.” ”T he plant root dehydrogenase activity were measured by using assay kits (Comin, Suzhou, China)” were added in the text.

Comment:L. 152. It is important for researchers to know the variety, not its number. Therefore, in tables 1, 3, 4, “serial number” should be replaced with the name of the variety.

Response: Thanks for your suggestions, we have replaced “serial number”with the name of the variety.

Comment: L. 169. Need a link to a figure or table.

Response: we are sorry for our negligence. We have linked it to table 1.

Comment: L. 178. «As shown in Table 3, HR plants under heat stress showed a higher chlorophyll level relative to HS cultivars with…..»

However, it is not clear from the table which varieties are HR and HS.

Response: we are sorry for providing vague information. We have improved Table 1 to show the HR and HS cultivars.

Comment:L. 181-182. «Previous study found that the HR varieties have a relatively higher chlorophyll content compared to HS varieties.»

It is not clear which previous studies are in question.

Response: thanks for your kindly advice, we have revised it as ” Previous study found that the leaf chlorophyll content was decreased under heat stress which further leads to a decreased sucrose content [16].”

Comment:L. 188. Chlorophyll a (r=-0.836) and Chlorophyll b (r=-0.797) are negatively correlated with the heat damage index,…. Where does this data come from? No reference to figure or table.

Response: We are sorry for our negligence. We have linked it to table 6.

Comment:L. 193. «From the Table 4, the SS content in the four varieties of Aoxia, Xinxiaqing, jiaoyang and liehuojingang were significantly higher, the SS content of HS varieties xinlvxiu…».

Table 4 does not contain data on the content of soluble sugars and proteins in leaves cabbage.

  1. 203. «The SP level of HR varieties is twice than that of HS varieties.»Where is this data?

Response: I'm sorry for making this mistake. We have revised the table 4 and delete “The SP level of HR varieties is twice than that of HS varieties.”

Comment: L. 208. «Our data showed that the heat damage index was highly negatively correlated with the content of SS (r=-0.86) and SP (r=-0.695).»These data are missing from Table 5.

Response: We are sorry for our negligence. We added these data in the table 6.

Comment:L. 217. «Heat stress initiaHeat sHeat stress initially….»

There is an error here.L. 218-220. Sentences duplicate each other - repetition.

Response: thanks for your careful review. We delete this sentence to avoid repetition.

Comment:L. 230. «In this work, the high temperature treatment was implemented on twenty B.rapa cultivars for 24 h.»

However, the Methods indicate the dynamics - from 1 to 7 days. It should be specified in the Methods.

Response:We are sorry for the mistake here. Thank you for your guidance. We have corrected it as” In this work, the high temperature treatment was implement on twenty B.rapa cultivars for 7 days. “

Comment: L. 223. «As shown in the Table 3, the MDA content was significantly accumulated in HS plants (aijiaohuang, xinlvxiu and wuyueman).»This is not visible from the table, because cultivar names are not given.The absence of variety names in tables 1, 3 and 4 makes it difficult to review.

Response: thanks for your advice. We have added cultivar names in the table 1,3,4,5 .

Comment: L. 308. «4.1. Biochemical analysis of enzymatic activities and Hydroxyl scavenging capacity, et al.»Why is there a Results section in the Discussion (Figures 2, 3, 4)?

Response:Thank you for your reasonable suggestion. We have move this part in to discussion section.

We appreciate for Reviewers’ warm work earnestly, and hope that the correction will meet with approval. Should you have any questions, please contact us without hesitate. 

Once again, thank you very much for your comments and suggestions.

Yours Sincerely,

Qingliang      Niu

Round 2

Reviewer 2 Report

The authors agreed with all my comments and wishes, and made all the edits to the text.